# Effective descriptions of bosonic systems can be considered complete

Francesco Arzani [1], Robert I. Booth [2,3] & Ulysse Chabaud [1] ✉

Bosonic statistics give rise to remarkable phenomena, from the Hong–Ou–Mandel effect to Bose–Einstein condensation, with applications spanning fundamental science to quantum technologies. Modeling bosonic systems relies heavily on effective descriptions: typically, truncating their infinite-dimensional state space or restricting their dynamics to a simple class of Hamiltonians, such as polynomials of canonical operators. However, many natural bosonic Hamiltonians do not belong to these simple classes, and some quantum effects harnessed by bosonic computers inherently require infinite-dimensional spaces. Can we trust that results obtained with such simplifying assumptions capture real effects? We solve this outstanding problem, showing that these effective descriptions do correctly capture the physics of bosonic systems. Our technical contributions are twofold: firstly, we prove that any physical bosonic unitary evolution can be accurately approximated by a finite-dimensional unitary evolution; secondly, we show that any finite-dimensional unitary evolution can be generated exactly by a bosonic Hamiltonian that is a polynomial of canonical operators. Beyond their fundamental significance, our results have implications for classical and quantum simulations of bosonic systems, provide universal methods for engineering bosonic quantum states and Hamiltonians, show that polynomial Hamiltonians generate universal gate sets for quantum computing over bosonic modes, and lead to a bosonic Solovay–Kitaev theorem.

Bosonic systems, often referred to as continuous-variable quantum systems in the context of quantum computing, encompass a number of promising setups for quantum information processing. These include photonic setups, which have led to some of the first demonstrations of quantum computational speedup[1–4]. Moreover, bosonic error-correcting codes[5], specifically those theorized in 2001 by Gottesman, Kitaev and Preskill (GKP)[6], have recently led to the first demonstration of quantum error-correction beyond the break-even point[7].

The description of bosonic systems requires a Hilbert space of infinite dimension, since it must support position and momentum operators satisfying the canonical commutation relation $[\hat{q}, \hat{p}] = i\hat{I}$ (such a relation cannot be satisfied by finite-dimensional linear operators, as can be seen by contradiction by taking the trace, giving zero for the left hand side and a non-zero value for the right hand side). While this large state space allows us to robustly encode quantum information as in GKP codes[6], it also makes modeling bosonic systems and computations challenging. This has motivated the use of mathematically simpler effective descriptions, aiming to capture their important features. Here we consider two widely adopted simplifications: effective dimension–simplifying the state space–and effective Hamiltonians–simplifying the dynamics.

Choosing an effective dimension for a bosonic system amounts to imposing a cut-off of its infinite-dimensional Hilbert space leading to a finite-dimensional one, based for instance on an energy bound. It allows us to simulate these systems on classical or quantum

[1]DIENS, École Normale Supérieure, PSL University, CNRS, INRIA, Paris, France. [2]University of Edinburgh, Edinburgh, UK. [3]University of Bristol, Bristol, UK.
✉e-mail: ulysse.chabaud@inria.fr

computers, up to a truncation error[8,9], and to draw equivalences with spin systems[10].

Alternatively, one may restrict to a specific class of effective Hamiltonians. A standard choice for bosonic computations is the set of polynomials in the canonical bosonic operators[11], which we refer to as polynomial Hamiltonians for brevity. Polynomial Hamiltonians are ubiquitous in physics, such as in the Bose–Hubbard model of interacting spinless bosons on a lattice[12], in the $\phi^4$ theory[13] in quantum field theory, or in the multipole expansion of the dielectric polarization in nonlinear optics[14], to a name a few. In particular, the standard model of quantum computations with continuous variables was defined by Lloyd and Braunstein based on unitary gates generated by polynomial Hamiltonians[11]. Furthermore, universality for continuous-variable quantum computing is defined as the ability of a gate set to approximate any polynomial Hamiltonian evolution[11,15].

While these effective descriptions greatly simplify the modeling of bosonic computations, they also have drawbacks: recall that descriptions based on a finite effective dimension cannot reproduce the bosonic canonical commutation relation exactly. In addition, there are fundamental differences between the sets of quantum correlations in finite and infinite-dimensional spaces[16–18]. These features suggest that some bosonic quantum effects require infinite-dimensional spaces and may not be accounted for when restricted to a finite effective dimension.

Similar doubts can be raised about the ability of effective Hamiltonians to faithfully describe bosonic phenomena, as many bosonic systems are not naturally described by polynomial Hamiltonians. For instance, Hamiltonians of Josephson junctions in superconducting systems include terms of the form $\cos\hat{\delta}$, where $\hat{\delta}$ is a canonical phase-difference operator[19]. Here, the function cos cannot be replaced by a polynomial–such as its truncated Taylor expansion–because the operator $\hat{\delta}$ is unbounded[20]. Moreover, in the context of bosonic computations, it is in fact not known whether the definition of universality, based on the ability to reproduce any evolution generated by polynomial Hamiltonians[11], is sound: by this we mean that it is far from obvious that universal continuous-variable quantum computers as defined by Lloyd and Braunstein satisfy plausible requirements that

one would expect by analogy with their discrete-variable counterparts. One example is universal state preparation, loosely defined as the ability to approximately prepare any quantum state from a given reference state: as far as we know, there could be inaccessible regions of the Hilbert space when one restricts to using gates generated by polynomial Hamiltonians. This can be phrased as a question about universal quantum control and was first formalized in this context as an open problem in ref. 21. Beyond this controllability problem, it is also highly non-trivial whether any physical unitary evolution may be approximated to arbitrary precision by a sequence of gates generated by polynomial Hamiltonians. By contrast, the definition of universality for discrete-variable quantum gate sets is based on the property that they allow us to approximate any unitary operation to arbitrary precision[22]. What is more, the Solovay–Kitaev theorem guarantees that these finite-dimensional universal gate sets provide efficient approximations of any unitary operator on a constant number of subsystems[23]. Such a theorem is missing in the general infinite-dimensional setting[24] and would be fundamental for understanding the computational complexity of bosonic systems, because it would imply that the complexity of bosonic computations is independent of the choice of gate set[25]. A pre-requisite for this result is to ensure that the definition of universality for continuous-variable quantum computing is sound: at the moment, it is unclear whether effective polynomial Hamiltonians provide a solid foundation for modeling bosonic computations.

Are these drawbacks simply mathematical artifacts, or should they be taken into account when describing bosonic systems? In other terms: Can effective descriptions of bosonic systems be considered complete?

Here we provide a positive answer to this question. As we shall demonstrate, this is not only a non-trivial conceptual question, but also a highly fruitful one that leads to several applications (see Fig. 1). Our technical contributions are twofold:

Firstly, we show that any physical unitary evolution can be approximated to arbitrary precision by a finite-dimensional evolution. The proof is based on an operationally meaningful and mathematically rigorous notion of approximation. Moreover, the corresponding cut-

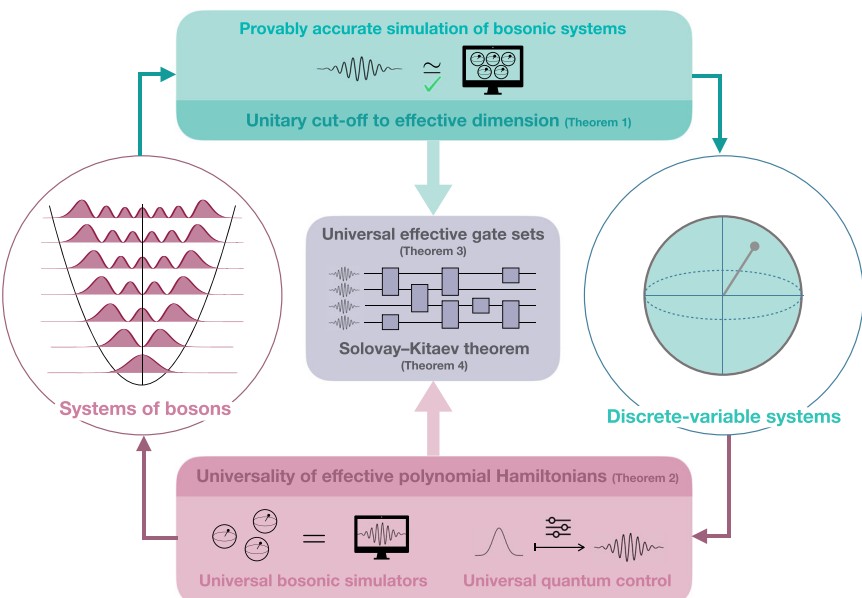

**Fig. 1 | Completeness of effective descriptions of bosonic systems.** Top: physical unitary dynamics of bosonic systems can be cut off to an effective dimension (Theorem 1), leading to rigorous precision guarantees for simulating these systems. Bottom: effective polynomial Hamiltonians can realize any finite-dimensional Hamiltonian (Theorem 2), enabling universal quantum control with these Hamiltonians and opening the way to universal bosonic simulators. Middle: combining the previous results, we design universal gate sets for bosonic computations based on effective polynomial Hamiltonians (Theorem 3), which satisfy a Solovay--Kitaev theorem (Theorem 4).

off unitary evolution and its effective dimension may be computed explicitly. In particular, the effective dimension relates to the amount of energy needed to implement the evolution, thus also yielding an algorithm for the simulation of bosonic dynamics on digital quantum computers with explicit space complexity (Theorem 1).

Secondly, we show that polynomial Hamiltonians can realize any finite-dimensional Hamiltonian exactly. We provide a constructive proof, based on Lagrange interpolation polynomials (Theorem 2). Together with the previous result, this shows that polynomial Hamiltonians can generate any physical unitary evolution approximately to arbitrary precision.

From a conceptual standpoint, our first technical result identifies a clear trade-off between space and energy when simulating bosonic systems, be it with classical or finite-dimensional quantum methods. Our second technical result provides universal methods for state and operator engineering using polynomial Hamiltonians, which can be used to simulate the dynamics of complex quantum systems using bosonic architectures. Furthermore, it resolves the open question about the universal quantum controllability of polynomial Hamiltonians in infinite dimension from ref.[21] (Corollary 1). Finally, these two technical contributions together show that any physical unitary evolution of a bosonic system may be approximated to arbitrary precision by a sequence of gates generated by polynomial Hamiltonians in a mathematically rigorous sense (Theorem 3). This solves the universality problem in continuous-variable quantum computing, showing that the definition of universality from ref.[11] is sound, and further leads to a Solovay–Kitaev theorem in infinite dimension for specific universal gate sets generated by polynomial Hamiltonians (Theorem 4).

## Results

### Bosonic quantum information theory

Quantum information theory with bosonic systems typically involves modeling unbounded operators over infinite-dimensional Hilbert spaces with continuous spectrum, and is thus often referred to as continuous-variable quantum information theory[26–28]. In this framework, systems corresponding to individual quantum harmonic oscillators are called qumodes, or simply modes, and quantum states of $m$-mode bosonic systems are elements of an infinite-dimensional Hilbert space $\mathcal{H}^{\otimes m}$, with each single-mode Hilbert space spanned by a countably infinite (Fock) basis $\{|n\rangle\}_{n\in\mathbb{N}}$.

Creation and annihilation operators $\hat{a}^\dagger$ and $\hat{a}$ for each mode are defined by their action on the Fock basis as $\hat{a}^\dagger|n\rangle = \sqrt{n+1}|n+1\rangle$, $\hat{a}|n+1\rangle = \sqrt{n+1}|n\rangle$ and $\hat{a}|0\rangle = 0$. The (unbounded) canonical bosonic operators are then given by $\hat{q} = \frac{1}{\sqrt{2}}(\hat{a}+\hat{a}^\dagger)$ and $\hat{p} = \frac{1}{i\sqrt{2}}(\hat{a}-\hat{a}^\dagger)$ and satisfy the canonical commutation relation $[\hat{q},\hat{p}] = i\hat{I}$, where $\hat{I}$ is the identity operator. Polynomial Hamiltonians over $m$ modes are the Hermitian operators of the form $P(\hat{q}_1,\hat{p}_1,\ldots,\hat{q}_m,\hat{p}_m)$, where $P$ is a polynomial and where $(\hat{q}_1,\hat{p}_1,\ldots,\hat{q}_m,\hat{p}_m)$ are the canonical operators of the modes $1,\ldots,m$. The number operator is defined as $\hat{n} = \hat{a}^\dagger\hat{a}$, and its expectation value is the average particle number of a bosonic mode, which we refer to as energy hereafter. For all $d\in\mathbb{N}$, we denote by $\mathcal{H}_d := \mathrm{span}\{|n\rangle\}_{0\le n\le d}$ the subspace of states with a number of particles at most $d$.

In some sense, not all states in the Hilbert space are valid physical states, even if they are normalized: for instance, some may have infinite energy, such as $\sqrt{6}\sum_{n\ge1}\frac{1}{\pi n}|n\rangle$. Similarly, there are normalized states with infinite fluctuations in either position or momentum or both. To avoid inconsistencies and ensure that the action of polynomial Hamiltonians leads to physical states, it is common to define a set of physical states as a dense subspace of the Hilbert space known as the Schwartz space[20]—informally, the set of states with bounded canonical operator moments, with a position wave function and all of its derivatives decaying sufficiently fast at infinity, which we denote by $\mathcal{S}\subset\mathcal{H}$ (see the Methods section). Similarly, denoting by $\mathcal{U}(\mathcal{H})$ the group of

unitary operators over $\mathcal{H}$, we define

$$\mathcal{U}(\mathcal{S}) := \left\{\hat{U}\in\mathcal{U}(\mathcal{H}) \mid \forall|\psi\rangle\in\mathcal{S}, \hat{U}|\psi\rangle\in\mathcal{S}\right\}. \tag{1}$$

These form a group of physical unitary operators, i.e., which map physical states to physical states. This set can be readily extended to the multimode setting. Physical unitary operators have the property that they map bosonic states with a finite number of particles to states of finite energy: for all $n\in\mathbb{N}$ and all $\hat{U}\in\mathcal{U}(\mathcal{S})$,

$$E_{\hat{U}}(n) := \sup_{|\psi\rangle\in\mathcal{H}_n} \langle\psi|\hat{U}^\dagger\hat{n}\hat{U}|\psi\rangle < +\infty. \tag{2}$$

In that case, the supremum is a maximum and the quantity $E_{\hat{U}}(n)$ can be thought of as the maximal amount of energy involved when implementing the unitary evolution $\hat{U}$ of an initial state with a number of particles at most $n$. This is a slightly weaker property than energy-limited quantum dynamics[29] which map state of finite energy to states of finite energy.

We denote by $D(\rho,\sigma) = \frac{1}{2}\|\rho-\sigma\|_1$ the trace norm distance between density operators[22], which is used to define the diamond norm for quantum maps. The diamond norm induces a distinguishability measure between maps that is often too stringent for quantum mechanical applications involving unbounded operators, as physical unitary channels may be always maximally separated for this norm, even though they may have similar effects on all states below a certain energy[30]. Instead, the strong topology of pointwise convergence provides a suitable notion of continuity in infinite dimensions, which is implied by the closeness in energy-constrained diamond norm (see the Methods section), defined as[31]:

$$\|\mathcal{E}\|_\diamond^E := \sup_{\mathrm{Tr}[\rho(\hat{H}\otimes\hat{I})]\le E} \|(\mathcal{E}\otimes\mathrm{id})\rho\|_1. \tag{3}$$

Here, $\mathcal{E}$ is a Hermitian-preserving map, $\rho$ is a density operator over $\mathcal{H}\otimes\mathcal{H}'$, with $\mathcal{H}'$ isomorphic to $\mathcal{H}$, and $\hat{H}$ is a Hamiltonian over $\mathcal{H}$ with ground state energy equal to 0 used to specify the energy bound $E > 0$. We employ this topology hereafter, with $\hat{H} = \hat{n}$ the number operator.

### Effective dimension of physical unitary evolutions

Having introduced the necessary notation, we show in this section that any physical unitary channel can be approximated to arbitrary precision by a finite-dimensional unitary channel, where the quality of the approximation is measured by the energy-constrained diamond norm. This is captured by the notion of approximate effective dimension, which we define as follows:

**Definition 1.** (Approximate effective dimension). Let $\hat{U}$ be a single-mode unitary operator, let $E\ge0$ be an energy parameter, and let $\epsilon > 0$ be an approximation parameter. Let also $d\in\mathbb{N}$. The unitary operator $\hat{U}$ has $(E,\epsilon)$-approximate effective dimension $d+1$ if there exists a unitary operator $\hat{V}_d$ over $\mathcal{H}_d$ which approximates $\hat{U}$ in the following sense: for any unitary operator $\hat{V}'$ over $\mathrm{span}(|n\rangle)_{n>d}$, denoting $\hat{V} = \hat{V}_d\oplus\hat{V}'$, $\mathcal{V} = \hat{V}\hat{V}^\dagger$ and $\mathcal{U} = \hat{U}\hat{U}^\dagger$,

$$\|\mathcal{U} - \mathcal{V}\|_\diamond^E \le \epsilon. \tag{4}$$

When that is the case, we say that $\hat{V}_d$ is a $(d+1)$-dimensional $(E,\epsilon)$-approximation of $\hat{U}$.

When a unitary dynamics has a finite approximate effective dimension, it can effectively be modeled by a finite-dimensional unitary evolution. We now give our first main result:

**Theorem 1.** (Effective dimension of physical unitary channels). Let $\mathcal{U} = \hat{U}\hat{U}^\dagger$ be a single-mode unitary channel, with $\hat{U}\in\mathcal{U}(\mathcal{S})$ a physical unitary operator. Let $E\ge0$ be an energy parameter, and let $\epsilon > 0$ be an approximation parameter. Then, there exists $d = \mathcal{O}\left(\frac{E}{\epsilon^4}E_{\hat{U}}\left(\frac{64E}{\epsilon^2}\right)\right)\in\mathbb{N}$

such that $\hat{U}$ has $(E, \epsilon)$-approximate effective dimension $d + 1$. Moreover, a finite-dimensional $(E, \epsilon)$-approximation of $\hat{U}$ can be computed in polynomial time in $d$, $E$, and $\frac{1}{\epsilon}$.

The proof of this result is based on constructing a good unitary operator approximation of a truncated unitary operator, while carefully controlling truncation parameters for the state space, before and after applying the unitary channel, so that the truncated unitary operator is itself a good approximation of the original unitary operator. We refer to the Methods section for the main techniques and to the Supplementary Information for the proof.

A direct consequence of Theorem 1 is that the corresponding sequence of $(E, \epsilon)$-approximations strongly converges to $\hat{U}$ as $\epsilon$ goes to 0 in trace distance (see the Methods section). From an operational standpoint, this implies that we can effectively truncate the Hilbert space of physical bosonic computations to a certain effective dimension, as long as the cut-off is high enough. As such, Theorem 1 provides rigorous bounds on the effective dimension required to simulate bosonic dynamics with provable accuracy[9]. In other words, there will be no catastrophic loss of information in the measurements statistics when restricting to the cut-off evolution: by the operational property of the trace distance, results of a cut-off bosonic computation will be indistinguishable from results of the original computation. However, this cut-off comes at a price: Theorem 1 identifies a direct relation between the effective dimension sufficient to embed a unitary evolution to a good precision and the energy $E_{\hat{U}}$ of the unitary evolution, as defined in Eq. (2). Such an energy bound may grow very fast for seemingly simple dynamics, such as alternating Gaussian and non-Gaussian unitary evolutions[25]. We give an explicit construction in the Supplementary Information showing that this growth may be arbitrarily fast.

## Universality of polynomial Hamiltonians

We have shown that bosonic unitary evolutions may be well approximated by finite-dimensional ones. In this section, we give our second main result, namely that any finite-dimensional Hamiltonian generating such a finite-dimensional unitary evolution can be realized exactly by a polynomial Hamiltonian:

**Theorem 2.** (Finite-dimensional universality of polynomial Hamiltonians). Let $d \in \mathbb{N}$ and let $\hat{H}$ be a Hermitian operator over $\mathcal{H}_d$. There exists a polynomial Hamiltonian $P_{\hat{H}}(\hat{q}, \hat{p})$ of degree at most $3d$ over $\mathcal{H}$ such that

$$P_{\hat{H}}(\hat{q}, \hat{p}) = \hat{H} \oplus \hat{H}', \tag{5}$$

where $\hat{H}'$ is a Hermitian operator over $\text{span}(|n\rangle)_{n > d}$. In particular, $P_{\hat{H}}(\hat{q}, \hat{p})$ generates an evolution in $\mathcal{H}$ given for all $|\psi\rangle \in \mathcal{H}_d$ by

$$e^{iP_{\hat{H}}(\hat{q}, \hat{p})}|\psi\rangle = e^{i\hat{H}}|\psi\rangle, \tag{6}$$

Moreover, the polynomial $P_{\hat{H}}$ can be computed in polynomial time in $d$.

The proof of this result is based on the use of interpolation polynomials for reproducing exactly a target finite-dimensional operator on a subspace of the Fock basis, taking advantage of the sparsity of the canonical operators in that basis. We refer to the Methods section for the main techniques and to the Supplementary Information for the multimode generalisation of Theorem 2 and its proof. In particular, defining the unitary evolution operator $e^{iP_{\hat{H}}(\hat{q}, \hat{p})}$ on $\mathcal{H}$ is highly non-trivial: we restrict its action to states in $\mathcal{H}_d$ to avoid such mathematical technicalities.

An important consequence of this theorem is that polynomial Hamiltonians allow us to explore the full Hilbert space, i.e., there are no inaccessible regions of the Hilbert space when starting from a fixed reference state, such as the vacuum state, and restricting to using gates generated by polynomial Hamiltonians:

**Corollary 1.** (Universal quantum controllability of polynomial Hamiltonians in infinite dimension). Let $|\psi\rangle \in \mathcal{H}$. For all $\epsilon > 0$, there exists a polynomial Hamiltonian $P_\epsilon(\hat{q}, \hat{p})$ such that evolving the vacuum state $|0\rangle$ under the Schrödinger equation with Hamiltonian $P_\epsilon(\hat{q}, \hat{p})$ for constant time yields a state $|\psi_\epsilon\rangle$ satisfying $D(|\psi\rangle, |\psi_\epsilon\rangle) \le \epsilon$.

Corollary 1 proves the universal quantum controllability of polynomial Hamiltonians in infinite dimensions, resolving the open question in[21] and [ref. 25, Open question 3]. We give a quick proof in the single-mode case hereafter, the multimode case being analogous.

**Proof.** Since the state $|\psi\rangle$ is normalized, there exists $d_\epsilon$ such it is $\epsilon$-close in trace distance to the state truncated at $d_\epsilon$ and renormalized, which we denote by $|\psi_\epsilon\rangle$. Then, there exists a finite-dimensional unitary operator $\hat{U}_\epsilon = e^{i\hat{H}_\epsilon}$ over $\mathcal{H}_{d_\epsilon}$ which maps $|0\rangle$ to $|\psi_\epsilon\rangle$. By Theorem 2, the polynomial Hamiltonial $P_\epsilon(\hat{q}, \hat{p}) := P_{\hat{H}_\epsilon}(\hat{q}, \hat{p})$ satisfies $P_\epsilon(\hat{q}, \hat{p}) = \hat{H}_\epsilon \oplus \hat{H}'_\epsilon$. Evolving the vacuum under the Schrödinger equation with Hamiltonian $P_\epsilon(\hat{q}, \hat{p})$ thus leads to the same state as if evolving the vacuum under the $(d_\epsilon + 1)$-dimensional Schrödinger equation with Hamiltonian $\hat{H}_\epsilon$, which concludes the proof. $\square$

Beyond quantum state preparation, Theorem 2 provides a universal method for exactly reproducing any finite-dimensional Hamiltonian using polynomial Hamiltonians. This method can be used to engineer universal bosonic simulators capable of emulating the unitary evolution of any discrete-variable quantum system.

Furthermore, Theorem 2 also implies that bosonic circuits composed of input vacuum state, unitary gates generated by polynomial Hamiltonians, and number measurements, can simulate universal qudit computations efficiently (without requiring feed-forward of measurement outcomes), generalizing a recent result for qubit computations [ref. 25, Theorem 4.1].

## Effective bosonic computations

In this section, we explore the combined consequences of Theorem 1 (effective dimensions) and Theorem 2 (effective Hamiltonians) for the validity of effective descriptions of bosonic computations. At a fundamental level, both results show that effective descriptions are capturing bosonic computations well, i.e., one can always truncate the dimension of physical unitary evolutions up to an arbitrarily small error and use polynomial Hamiltonians to model finite-dimensional evolutions exactly. Together, these two results imply that polynomial Hamiltonians can generate any physical unitary evolution approximately to arbitrary precision:

**Theorem 3.** (Infinite-dimensional universality of polynomial Hamiltonians). Let $\mathcal{U} = \hat{U}\hat{U}^\dagger$ be a single-mode unitary channel, with $\hat{U} \in \mathcal{U}(\mathcal{S})$ a physical unitary operator. Let $E \ge 0$ be an energy parameter, and let $\epsilon > 0$ be an approximation parameter. Then, there exists $d = \mathcal{O}\left(\frac{E}{\epsilon^4} E_{\hat{U}}\left(\frac{64E}{\epsilon^2}\right)\right) \in \mathbb{N}$ and a polynomial Hamiltonian $P(\hat{q}, \hat{p})$ of degree at most $3d$ such that, writing $\mathcal{V}$ the corresponding unitary evolution induced on $\mathcal{H}_d$,

$$\| \mathcal{U} - \mathcal{V} \|^E_\diamond \le \epsilon. \tag{7}$$

Moreover, the polynomial $P$ can be computed from $\hat{U}$ in polynomial time in $d$, $E$, and $\frac{1}{\epsilon}$.

As we explain in the Methods section, this result is a direct consequence of Theorem 1 and Theorem 2. Formally, Theorem 3 shows that the group of unitary gates generated by polynomial Hamiltonians is dense in the group of physical unitary operators in the strong operator topology (see the Methods section). This places the definition of universality for continuous-variable quantum gate sets, based on the ability to approximate to arbitrary precision any unitary

evolution generated by a polynomial Hamiltonian[11], on an equal footing with its discrete-variable counterpart, based on the property to approximate to arbitrary precision any unitary operation[22].

As such, this shows that polynomial Hamiltonians generate truly universal gate sets and paves the way for a Solovay–Kitaev theorem[23] in the infinite-dimensional setting. We obtain a version of this theorem hereafter, for universal gate sets generated by polynomial Hamiltonians as in Theorem 2:

**Theorem 4.** (Solovay–Kitaev theorem for polynomial Hamiltonians). Let $E > 0$, $\epsilon > 0$ and $d \geq \frac{64E}{\epsilon^2} \in \mathbb{N}$. Let $\mathcal{G}$ be a finite set of unitary operators over $\mathcal{H}_d$ generating a dense subset of $\mathcal{U}(\mathcal{H}_d)$ and let $\mathcal{P}$ be its realization with polynomial Hamiltonians from Theorem 2. There is a constant $c$ such that for any physical unitary operator $\hat{U} \in \mathcal{U}(\mathcal{S})$ with $(E, \epsilon)$-approximate effective dimension $d + 1$, there exists a finite sequence $\hat{V}$ of gates from $\mathcal{P}$ of length $\mathcal{O}(\log^c(1/\epsilon))$ and such that $\| \mathcal{U} - \mathcal{V} \|_\diamond^E \leq 2\epsilon$, where $\mathcal{U} = \hat{U} \hat{U}^\dagger$ and $\mathcal{V} = \hat{V} \hat{V}^\dagger$.

Recall that Theorem 1 ensures the existence of an $(E, \epsilon)$-approximate effective dimension for all physical unitary operators. The proof of Theorem 4 combines Theorems 1 and 2 with the Solovay–Kitaev theorem for qudits[23] of dimension $d + 1$, and provides an explicit algorithm for producing the sequence of polynomial Hamiltonians, given the target unitary $\hat{U} \in \mathcal{U}(\mathcal{S})$. We refer to the Methods section for the main techniques and to the Supplementary Information for the proof.

Theorem 4 generalises the Gaussian Solovay–Kitaev theorem from ref. 24, thus providing an answer to [25, Open question 4] for the gate sets based on the polynomial Hamiltonians used in Theorem 2. In particular, it implies the computational equivalence of bosonic quantum circuits based on these universal gate sets.

## Discussion
We have considered effective descriptions of bosonic systems based on effective dimension and effective (polynomial) Hamiltonians, and rigorously proved that they faithfully reproduce the features of the original dynamics. Our findings have several far-reaching consequences for the simulation of bosonic systems and the complexity of bosonic computations, summarised in Fig. 1: (i) it is always possible to truncate the infinite-dimensional Hilbert space of physical bosonic unitary dynamics to an effective dimension depending on the energy involved, with negligible loss of information (Theorem 1), leading to explicit bounds on the space complexity of the simulation of bosonic systems; (ii) polynomial Hamiltonians provide universal bosonic simulators of finite-dimensional quantum systems (Theorem 2) and enable universal bosonic quantum state engineering (Corollary 1), which solves a long-standing open problem on the universal quantum controllability of polynomial Hamiltonians[21]; (iii) polynomial Hamiltonians generate universal gate sets in infinite dimensions (Theorem 3), which is of foundational importance for continuous-variable quantum computing, since it proves the validity of the definition of universality based on polynomial Hamiltonians. This was left open since the seminal paper of Lloyd and Braunstein[11], and places it on an equal footing with its discrete-variable counterpart; (iv) these universal gate sets satisfy an infinite-dimensional Solovay–Kitaev theorem, leading to bosonic circuits that are equivalent from a computational complexity standpoint (Theorem 4), thus providing a solid foundation for the complexity theory of bosonic computations[25].

Our results also motivate several new research directions. For instance, we expect that our Theorem 1 on effective dimensions of infinite-dimensional unitary dynamics may be generalized to multimode, non-unitary dynamics, as a consequence of Stinespring's dilation theorem[22,32].

Moreover, Theorem 2 together with Corollary 1 on the universality of polynomial Hamiltonians may enable the experimental implementation of new quantum simulators[33] and the preparation of new exotic quantum states using bosonic platforms, including superconducting[34] and photonic[35] setups. It is also an intriguing question whether the universality of polynomial Hamiltonians can be extended to hybrid boson-fermion and boson-qubit systems[36].

On a different note, it has been argued in several works[11,15,21] that Gaussian Hamiltonians (of degree $\leq 2$) together with any single non-Gaussian polynomial Hamiltonian (of degree $> 2$) are sufficient to generate any polynomial Hamiltonian evolution approximately. However, a formal proof of this statement is missing, although Trotter–Suzuki formulas with rigorous error bounds have been recently uncovered in the infinite-dimensional setting[37]. Combined with our results, formalizing these polynomial gate set equivalences could lead to versions of Theorem 2 and Theorem 3 valid for any choice of non-Gaussian polynomial Hamiltonian. In particular, this may unlock an infinite-dimensional Solovay–Kitaev for all polynomial gate sets rather than only those used in our Theorem 2.

In this regard, it is well-known that the complexity of the qudit Solovay–Kitaev algorithm for finding an approximating gate sequence scales poorly with the qudit dimension[23]. This implies in turn that our Solovay–Kitaev theorem (Theorem 4) has high complexity based on the effective dimension of the target physical unitary operator. It would be interesting to develop a version achieving better efficiency in terms of energy.

In summary, we have shown that effective descriptions are sufficient to capture and reproduce bosonic evolutions, with significant implications for quantum simulation, quantum state engineering, and universal quantum computing. From a conceptual standpoint, our work indicates that bosonic phenomena which cannot be reproduced, even approximately, in finite dimensions must involve unphysical states of infinite energy. Beyond the framework of bosonic systems, our work supports the view that there is much to be learned by challenging the assumptions underlying our idealized models of physical reality[38].

## Methods
### Preliminary material
**CV quantum information.** In continuous-variable quantum information[26–28], information is encoded in the state of $m$ bosonic modes, which are mathematically modelled by a Hilbert space $\mathcal{H}$ spanned by a countably-infinite Fock basis $\{|n\rangle\}_{n\in\mathbb{N}}$, i.e., $\sum_{n\in\mathbb{N}} c_n |n\rangle \in \mathcal{H}$ if and only if $\sum_{n\in\mathbb{N}} |c_n|^2 < \infty$.

To a single bosonic mode, we associate the annihilation operator $\hat{a}|n\rangle = \sqrt{n}|n-1\rangle$ and creation operator $\hat{a}^\dagger|n\rangle = \sqrt{n+1}|n+1\rangle$ which satisfy $[\hat{a}, \hat{a}^\dagger] = \hat{I}$. The canonical operators (also called position and momentum or quadrature operators) are defined as $\hat{q} = (\hat{a} + \hat{a}^\dagger)/\sqrt{2}$ and $\hat{p} = (\hat{a} - \hat{a}^\dagger)/i\sqrt{2}$, respectively.

Polynomials in the canonical operators are central in the theory of quantum computation over continuous variables, as introduced by Lloyd and Braunstein[11]. In particular, a universal continuous-variable quantum computer is defined as a device that can approximately generate any evolution over multimode bosonic systems that can be written as a (Hermitian) polynomial $P(\hat{q}_1, \hat{q}_2, \ldots, \hat{p}_1, \hat{p}_2, \ldots)$. In[11], an informal inductive reasoning is used to argue that arbitrary polynomials can be obtained though judicious commutation of all Hamiltonians of degree less than or equal to two and any Hamiltonian of higher order. The former generate so-called Gaussian transformations[26].

Although it is a mathematically convenient setting for analysing bosonic systems, the Hilbert space $\mathcal{H}$ has some drawbacks from a physical perspective. For instance, it contains states of infinite energy and states for which the position and momentum operators are not well-defined[20]. The *Schwartz space* $\mathcal{S}$ is defined as the dense subspace of elements $\sum_{n\in\mathbb{N}} c_n |n\rangle \in \mathcal{H}$ such that $\lim_{n\to 0} c_n n^k = 0$ for any natural number $k$[39]. The Schwartz space is not a Hilbert space, but it retains many of the useful analytical properties of $\mathcal{H}$ whilst eliminating these physical inconsistencies: every state in $\mathcal{S}$ has bounded energy, and

there is a well-defined action of polynomials of the canonical operators on those states[21]. Now, there are unitaries that map physical (Schwartz) states to non-physical states (e.g., with infinite energy), so we exclude these unitaries and allow only those that map physical states to physical states. Letting $\mathcal{U}(\mathcal{H})$ be the group of unitaries on $\mathcal{H}$, the group of physical unitaries is defined as

$$\mathcal{U}(\mathcal{S}) := \{\hat{U} \in \mathcal{U}(\mathcal{H}) | \forall |\psi\rangle \in \mathcal{S} : \hat{U}|\psi\rangle \in \mathcal{S}\}. \quad (8)$$

The elements of this group are closely related to the notion of Schwartz operators defined in[40]. We note that, in fact, all our results involving physical unitary operators (Theorems 1, 3 and 4) are valid under the weaker condition $\langle n|\hat{U}^\dagger \hat{n}\hat{U}|n\rangle < +\infty$ for all $n \in \mathbb{N}$, rather than $\hat{U} \in \mathcal{U}(\mathcal{S})$.

**Distance measures.** In making claims about approximating operators on $\mathcal{H}$ or $\mathcal{S}$, we must be explicit about which distance metric we are using to measure the quality of our approximations. The diamond norm for completely-positive channels is a natural candidate. It is defined as

$$||\mathcal{E}||_\diamond := \sup\{||(\mathcal{E} \otimes \mathrm{id})\rho||_1 | \rho \text{ density operator on } \mathcal{H} \otimes \mathcal{H}\},$$

where $|| \cdot ||_1$ denotes the trace norm. However, it has some undesirable properties: for instance, if $\hat{H}$ is any unbounded Hermitian operator, and $\mathcal{U}_{H,s} : \rho \mapsto e^{isH}\rho e^{-isH}$ is the associated one-parameter family of unitary channels, then there are density operators $\rho$ such that $||\mathcal{U}_{H,s}(\rho) - \mathcal{U}_{H,s+\epsilon}(\rho)||_1$ does not go to 0 even as $\epsilon$ does, so that the diamond norm $||\mathcal{U}_{H,s} - \mathcal{U}_{H,s+\epsilon}||_\diamond$ also does not go to 0 [ref. 30, Proposition 2]. This clearly precludes the use of the diamond norm for measuring the distance between two Hamiltonian evolutions in operational settings.

Instead, the energy-constrained diamond norms give a family of distance measures which avoid this pitfall[30,31]. They are defined by restricting the supremum to states whose energy is bounded by a parameter $E > 0$:

$$\| \mathcal{E}\|_\diamond^E : = \sup_{\mathrm{Tr}[\rho(\hat{H}\otimes\hat{I})] \le E} \| (\mathcal{E} \otimes \mathrm{id})\rho\|_1, \quad (9)$$

where now $\hat{H}$ is a reference energy operator. Mathematically, the energy-constrained diamond norms are related to the *strong operator topology*, which gives the correct notion of pointwise continuity in time for unitary evolutions:

**Proposition 1.** (ref. 31, Proposition 3). If, for any $E > 0$ and quantum channel $\mathcal{E}$, the sequence of quantum channels $\{\mathcal{E}_p\}_{p\in\mathbb{N}}$ is such that $\lim_{p\to\infty}||\mathcal{E}_p - \mathcal{E}||_\diamond^E = 0$ then $\{\mathcal{E}_p\}_{p\in\mathbb{N}}$ converges to $\mathcal{E}$ in the strong operator topology.

## Tools for proving Theorem 1
Theorem 1 is concerned with the first kind of effective description we consider, namely replacing an infinite-dimensional unitary operator $\hat{U}$ with a finite-dimensional approximation, while precisely quantifying the error introduced in the process.

The proof hinges on two types of approximations: firstly, we show that the error measured using an energy-constrained diamond norm can be bounded by considering only a truncated Fock space, introducing a suitable truncation parameter $M$; secondly, we use the physicality of the channel to focus on its action on a larger finite-dimensional subspace, introducing another suitable cut-off parameter $N \geq M$ in Fock space. The error introduced in both steps can be bounded making use of the so-called gentle measurement lemma:

**Lemma 1.** (Gentle measurement lemma[41], Lemma 9.4.2). Consider a density operator $\rho$ and a projector $\Pi$ where $0 \le \Pi \le \mathbb{I}$. Suppose that

$\mathrm{Tr}(\Pi\rho) \ge 1 - \epsilon, \quad \epsilon \in [0,1]$. Then,

$$||\rho - \Pi\rho\Pi||_1 \le 2\sqrt{\epsilon}. \quad (10)$$

Note that this is a slightly weaker result than the more general version in ref. 41, but it is sufficient for our proof. Earlier proofs of essentially the same result can be found in[42,43]. Furthermore, while we will use this version for the sake of simplicity, a slightly tighter bound was recently proven[44] (see Lemma 6 therein). We use this lemma twice, for the projectors $\Pi_M$ and $\Pi_N$ onto the subspace of states with a number of particles at most $M$ and $N$, respectively. Now, the restriction $\Pi_N\hat{U}\Pi_N$ of a unitary operator to a finite-dimensional subspace is not necessarily unitary. Therefore we further need to rely on a notion of closest unitary operator in order to construct a suitable unitary version of the cut-off of the original channel. This notion is provided by the QR factorization (related to the Gram–Schmidt process):

**Lemma 2.** (QR factorization for rectangular complex matrices). Let $A \in \mathbb{C}^{N\times M}$ be a matrix with linearly independent columns. Then, there exists a unitary matrix $Q \in \mathbb{C}^{N\times N}$ and an upper triangular matrix $R \in \mathbb{C}^{M\times M}$ such that:

$$A = Q\begin{pmatrix} R \\ 0 \end{pmatrix}. \quad (11)$$

We apply this decomposition to a column full rank submatrix of the cut-off operator $\Pi_N\hat{U}\Pi_N$ and choose the resulting $Q$ as our finite-dimensional unitary approximation of $\hat{U}$, carefully bounding the error introduced. Combining the approximation errors at each step completes the proof, yielding an effective dimension $d = N$. The complexity of computing the QR factorization (see for example[45], Section 5.2) for the cut-off dimension introduced determines the complexity of constructing the unitary cut-off version of the original channel, i.e., computing its effective description. We refer to the Supplementary Information for a detailed proof.

## Tools for proving Theorem 2
Theorem 2 seeks to reproduce a finite-dimensional unitary operator $\hat{U} = e^{iH}$ with a unitary operator generated by a polynomial Hamiltonian.

Since we are specifically interested in Hamiltonian evolutions where the Hamiltonian is an *unbounded* operator, we must unavoidably consider some of the mathematical technicalities in manipulating such operators. In particular, if $\hat{H}$ is an unbounded Hamiltonian, we cannot expect the usual power series expansion of the exponential, $e^{it\hat{H}} = \sum_{n\in\mathbb{N}} \frac{(it\hat{H})^n}{n!}$, to converge to an operator on $\mathcal{H}$. For sufficiently well-behaved unbounded operators, the exponential may be defined instead via the spectral theorem or the resolvent formalism[20,46]. However, there are still many cases in which defining unambiguously the unitary operator generated by a polynomial Hamiltonian is challenging, because the corresponding Schrödinger equation does not have solutions for most choices of initial conditions[20].

In the present case, we avoid these mathematical technicalities by focussing on polynomial Hamiltonians that are block-diagonal in the Fock basis, i.e., which take the form $P(\hat{q}, \hat{p}) = \Pi_N\hat{H}\Pi_N \oplus (I - \Pi_N)\hat{H}'(I - \Pi_N)$ for some $N \geq 0$, where $\Pi_N$ is the projector onto the subspace of states with a number of particles at most $N$. This ensures that, given an initial state with at most $N$ particles, the Schrödinger evolution with Hamiltonian $P(\hat{q}, \hat{p})$ remains in the subspace of states with a number of particles at most $N$, and reproduces the finite-dimensional unitary evolution $e^{i\Pi_N\hat{H}\Pi_N}$ on that subspace.

The proof of Theorem 2 then proceeds by constructing such a block-diagonal polynomial Hamiltonian $P(\hat{q}, \hat{p}) = \Pi_N\hat{H}\Pi_N \oplus (I - \Pi_N)\hat{H}'(I - \Pi_N)$, given the description of a finite-dimensional Hermitian operator $\hat{H}$. This construction is based on decomposing the finite-dimensional Hamiltonian $\hat{H}$ as a sum of terms pairing only two basis

states, say $|j\rangle$ and $|k\rangle$. These terms can be mimicked by polynomial Hamiltonians that are block-diagonal in the Fock basis, constructed using the canonical operators and polynomial functions that are zero on all integers in a specific range, except for the relevant ones, depending on $j$ and $k$. This is achieved using Lagrange interpolation polynomials:

**Definition 2.** (Lagrange interpolation polynomials). Given $n + 1$ pairs $(x_0, y_0), ..., (x_n, y_n)$, the Lagrange interpolation polynomial is defined as

$$L(x) = \sum_{j=0}^{n} y_j \prod_{\substack{0 \le i \le n \\ i \ne j}} \frac{x - x_i}{x_j - x_i}. \tag{12}$$

It satisfies $L(x_i) = y_i$, for all $0 \le i \le n$.

In particular, polynomial operators of the form $P(\hat{a}^\dagger \hat{a})$ with $P$ a well-chosen Lagrange interpolation polynomial allow us to reproduce diagonal entries of $\hat{H}$, while operators of the form $P(\hat{a}^\dagger \hat{a})\hat{a}^k + h.c.$ lead to off-diagonal entries. We refer to the Supplementary Information for a detailed proof, including a generalisation of the result to the multimode case.

## Proof of Theorem 3

Theorem 3 deals with the second kind of effective description we consider, namely approximating a physical infinite-dimensional unitary operator $\hat{U}$ with a unitary approximation generated by a polynomial Hamiltonian, while precisely quantifying the error introduced in the process.

**Proof.** The proof combines Theorem 1 and Theorem 2 as follows. We first use Theorem 1 to derive a $(d + 1)$-dimensional $(E, \epsilon)$-approximation $\hat{V}_d$ of the original unitary operator $\hat{U}$, with $d = \mathcal{O}\left(\frac{E}{\epsilon^4} E_{\hat{U}}\left(\frac{64E}{\epsilon^2}\right)\right)$, in time polynomial in $d$, $E$, and $\frac{1}{\epsilon}$. Then, we extract a finite-dimensional Hamiltonian $\hat{H}$ generating the unitary evolution $\hat{V}_d$. We explain hereafter how this step can be performed efficiently. Finally, we use Theorem 2 to construct a polynomial Hamiltonian of degree at most $3d$ which generates a unitary evolution that reproduces the unitary operator $\hat{V}_d = e^{iH}$, itself approximating the original unitary operator $\hat{U}$.

Since $\hat{V}_d \in \mathcal{U}(\mathcal{H}_d)$ is a unitary operator on a $(d + 1)$-dimensional Hilbert space, we can write $\hat{V}_d = WDW^\dagger$ for some unitary matrix $W$ and a diagonal matrix $D$, whose diagonal elements are of the form $(e^{i\lambda_n})_{n=0,...,d}$ with $\lambda_n \in [0, 2\pi)$. Let $\Sigma$ be the diagonal matrix whose diagonal elements are $(\lambda_n)_{n=0,...,d}$, then $\Sigma$ is Hermitian and $\hat{V}_d = We^{i\Sigma}W^\dagger = e^{iW\Sigma W^\dagger}$. Since diagonalising $D$ and the matrix multiplications with $\Sigma$ can both be performed in time polynomial in $d$, we can extract the generator $\hat{H} = W\Sigma W^\dagger$ of $\hat{V}_d$ efficiently, which concludes the proof of Theorem 3. $\square$

## Tools for proving Theorem 4

Theorem 4 provides a Solovay–Kitaev theorem for universal unitary gate sets generated by block-diagonal polynomial Hamiltonians constructed in Theorem 2.

Given a good finite-dimensional unitary approximation $\hat{V}_d$ of the physical unitary $\hat{U}$, guaranteed to exist by Theorem 1, the proof proceeds by synthesizing a short sequence of finite-dimensional gates approximating $\hat{V}_d$ over $\mathcal{H}_d$, via the Solovay–Kitaev theorem for qudits:

**Theorem 5.** (Qudit Solovay–Kitaev theorem[23]). Let $\mathcal{G}$ be a finite set of unitary gates with determinant one and their inverses, generating a dense subset of $SU(d)$. There is a constant $c$ such that for any $U \in SU(d)$, there exists a finite sequence $S$ of gates from $\mathcal{G}$ of length $\mathcal{O}(\log^c(\frac{1}{\epsilon}))$ and

such that $d(U, S) < \epsilon$, where $d$ is the distance induced by the operator norm.

We then employ Theorem 2 to generate each finite-dimensional unitary gate in the sequence using polynomial Hamiltonians, obtaining a short sequence of unitary operators over $\mathcal{H}$ generated by polynomial Hamiltonians approximation $\hat{U}$ over $\mathcal{H}_d$. Finally, we leverage the physicality of the target unitary channel to lift the approximation of $\hat{U}$ over $\mathcal{H}_d$ to an approximation over $\mathcal{H}$. We refer to the Supplementary Information for a detailed proof.

## Data availability

No data has been generated in this work.

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

## Acknowledgements

The authors warmly acknowledge inspiring discussions with F. Grosshans and D. Markham. F.A. is grateful to J. Eisert for insightful discussions. U.C. acknowledges stimulating discussions with S. Mehraban, M. Rørdam and P.-E. Emeriau. The authors also thank S. Mehraban for helpful comments on a previous version of this manuscript. U.C. is grateful to Á. Capel, N. Datta, L. Lami and A. Winter for organising the BIRS-IMAG workshop Towards infinite dimension and beyond in quantum information (May 2024, Granada, Spain), where open questions resolved in this work were discussed. We acknowledge funding from the Plan France 2030 project NISQ2LSQ (ANR-22-PETQ-0006). U.C. acknowledges funding from the European Union's Horizon Europe Framework Programme (EIC Pathfinder Challenge project Veriqub) under Grant Agreement No. 101114899.

## Author contributions

All authors (F.A., R.I.B. and U.C.) developed the theoretical framework, discussed the results and contributed to the preparation of the manuscript.

## Competing interests

The authors declare no competing interests.
