## [Transparent Peer Review file · Nature Communications]

Effective descriptions of bosonic systems can be considered complete

Corresponding Author: Dr Ulysse Chabaud

Version 0:

Reviewer comments:

Reviewer #1

(Remarks to the Author)

This work addresses a long-standing question in quantum information theory with continuous-variable systems: can the effective, often finite-dimensional, models typically used to describe bosonic systems be regarded as complete representations of the underlying infinite-dimensional physics? The authors answer this question affirmatively by establishing two main results (Theorems 1 and 2), together with several important corollaries (Theorems 3 and 4, Corollary 1).

In Theorem 1 they show that any physical unitary evolution of a bosonic system—formally, any unitary preserving the Schwartz space of physical states—can be approximated to arbitrary precision, in the energy-constrained diamond norm sense, by a unitary evolution acting on some finite-dimensional subspace spanned by the first $d+1$ Fock states. The authors also provide a constructive procedure to obtain such finite-dimensional approximations and quantify the required effective dimension in terms of the energy involved in the unitary implementation. This result places the simulation of bosonic dynamics on rigorous footing, showing that it is possible to truncate infinite-dimensional Hilbert spaces without compromising physical accuracy.

The second main contribution (Theorem 2) establishes that any finite-dimensional Hamiltonian can be exactly simulated by a Hamiltonian that is a polynomial of the canonical operators. This is achieved via an explicit construction based on Lagrange interpolation in the Fock basis. It immediately implies that polynomial Hamiltonians can prepare any state approximately starting from any fixed fiducial state (Corollary 1); this result has important implications for the controllability of continuous-variable quantum systems. By combining this result with the first, the authors show that arbitrary physical bosonic unitaries can be approximated arbitrarily well in the topology of strong convergence using evolutions generated by polynomial Hamiltonians (Theorem 3). This yields a rigorous justification for the standard notion of universality in continuous-variable quantum computing à la Lloyd-Braunstein. The last main result (Theorem 4) is a version of the Solovay-Kitaev theorem for bosonic systems, which applies to gate sets comprising polynomially generated unitaries.

I regard this work as a technically solid and conceptually important contribution to the field of continuous-variable quantum information and computation, and have no difficulty recommending publication, provided that some concerns I have are adequately addressed. I went through the whole paper and part of the SI, verifying all the main conceptual steps of the proofs. I reproduced some (but not all) calculations, and I am relatively confident they are correct, at least in essence.

Some further comments are below:

- Believing that in English one can write “these mathematical quantities *verify* this equation” is a very common mistake, especially among speakers of Romance languages. Alas, in English only people can *verify* equations; mathematical quantities instead *satisfy* them.

- The paper

M. Keyl, J. Kiukas, and R. F. Werner. Schwartz operators, *Reviews in Mathematical Physics* Vol. 28, No. 03, 1630001 (2016)

should be cited prominently, as it formalizes the notion of Schwartz operators widely employed in this work.

- p. 3: "This is a slightly weaker property than energy limited quantum dynamics". Is it not a slightly *stronger* property, in the sense that any evolution that satisfies (2) is also energy limited, but not vice versa?

- Ref. [32] has a published version, which should be cited.

- For an important result such as the gentle measurement lemma, a quick citation to a textbook such as [41] is in my view not appropriate. One should spend some time to trace back the contribution to the original works, which in this case are Lemma 4.3 in

E. B. Davies. Quantum stochastic processes, Communications in Mathematical Physics 15(4), 277-304 (1969),

whose proof contains essentially the statement, even in its tight form, as well as the better known

A. Winter. Coding theorem and strong converse for quantum channels, IEEE Transactions on Information Theory 45(7), 2481-2485 (1999).

- "notation" is uncountable; thus, "notations" does not exist in English.

- SI, p.2: "we now look for \hat{V} such that $\|(\mathcal{U}-\mathcal{V})(\phi)\|_1$ is uniformly bounded". It is for sure uniformly bounded, by the constant 2.

- The authors might be interested in knowing that Lemma 1 is not tight. A tight version has been recently proposed, under the name "gentler measurement lemma" as Lemma 6 (and, more precisely, Eq. (44)) in <https://arxiv.org/abs/2501.12447>.

- Lemma 2 in the SI is of course part of the folklore of quantum information, although I have to admit that a quick search did not reveal any precise pointers. A one-line proof uses the formula $\|X\|_1^2 = 2\text{Tr}[X^2] - (\text{Tr}[X])^2$, valid for any 2x2 matrix X with one positive and one negative eigenvalue.

- A general comment: I recall that a discussion on the open question tackled in this work took place at the conference "Towards Infinite Dimension and Beyond in Quantum Information", held in Granada in May 2023. It would have been nice to acknowledge the contribution of the attendees of that conference for the development of the present work, if any.

- p.5, bottom of 1st column: there are quite a few repetitions that diminish the quality of the prose; my tally is engineer(...) x3 and simulat(...) x2.

Reviewer #2

(Remarks to the Author)

Please find my report attached in the pdf file. At this point, I recommend acceptance after minor revision, as this paper resolves key conceptual and technical questions in CV quantum theory in very precise yet conceptually clear manner.

Version 1:

Reviewer comments:

Reviewer #2

(Remarks to the Author)

I have no further comments. The authors thoroughly addressed my concerns.

NCOMMS-25-28662-T: reply to reviewers

Francesco Arzani, Robert Booth, Ulysse Chabaud

We would like to sincerely thank the editor and reviewers for their careful reading of our manuscript and for the constructive comments provided. We appreciate the time and effort invested in reviewing our work and acknowledge the valuable suggestions that have helped us improve the manuscript.

We believe we have addressed all the issues raised in detail in the following point-by-point response. For each comment, we first restate the concern raised by the reviewers, then provide our response, and finally describe the specific changes made to the manuscript to address the comment.

We trust that the revised manuscript now meets the standards required for publication and look forward to your further feedback.

REVIEWER # 1

COMMENT # 1.1

This work addresses a long-standing question in quantum information theory with continuous-variable systems: can the effective, often finite-dimensional, models typically used to describe bosonic systems be regarded as complete representations of the underlying infinite-dimensional physics? The authors answer this question affirmatively by establishing two main results (Theorems 1 and 2), together with several important corollaries (Theorems 3 and 4, Corollary 1).

In Theorem 1 they show that any physical unitary evolution of a bosonic system—formally, any unitary preserving the Schwartz space of physical states—can be approximated to arbitrary precision, in the energy-constrained diamond norm sense, by a unitary evolution acting on some finite-dimensional subspace spanned by the first $d+1$ Fock states. The authors also provide a constructive procedure to obtain such finite-dimensional approximations and quantify the required effective dimension in terms of the energy involved in the unitary implementation. This result places the simulation of bosonic dynamics on rigorous footing, showing that it is possible to truncate infinite-dimensional Hilbert spaces without compromising physical accuracy.

The second main contribution (Theorem 2) establishes that any finite-dimensional Hamiltonian can be exactly simulated by a Hamiltonian that is a polynomial of the canonical operators. This is achieved via an explicit construction based on Lagrange interpolation in the Fock basis. It immediately implies that polynomial Hamiltonians can prepare any state approximately starting from any fixed fiducial state (Corollary 1); this result has important implications for the controllability of continuous-variable quantum systems. By combining this result with the first, the authors show that arbitrary physical bosonic unitaries can be approximated arbitrarily well in the topology of strong convergence using evolutions generated by polynomial Hamiltonians (Theorem 3). This yields a rigorous justification for the standard notion of universality in continuous-variable quantum computing à la Lloyd-Braunstein. The last main result (Theorem 4) is a version of the Solovay-Kitaev theorem for bosonic systems, which applies to gate sets comprising polynomially generated unitaries.

I regard this work as a technically solid and conceptually important contribution to the field of continuous-variable quantum information and computation, and have no difficulty recommending publication, provided that some concerns I have are adequately addressed. I went through the whole paper and part of the SI, verifying all the main conceptual steps of the proofs. I reproduced some (but not all) calculations, and I am relatively confident they are correct, at least in essence.

Some further comments are below:

Reply:

We thank the referee for taking the time to read our manuscript carefully and we are grateful for recommending it for publication. We address their comments in the following.

COMMENT # 1.2

*Believing that in English one can write “these mathematical quantities *verify* this equation” is a very common mistake, especially among speakers of Romance languages. Alas, in English only people can *verify* equations; mathematical quantities instead *satisfy* them.*

Reply:

We thank the referee for spotting this mistake, we have corrected it in the revised version.

COMMENT # 1.3

*The paper
M. Keyl, J. Kiukas, and R. F. Werner. Schwartz operators, Reviews in Mathematical Physics
Vol. 28, No. 03, 1630001 (2016)
should be cited prominently, as it formalizes the notion of Schwartz operators widely employed
in this work.*

Reply:

We thank the referee for their suggestion. The suggested reference now appears as reference [41] in the revised manuscript.

Changes:

We have added the following sentence after Eq. 8:

The elements of this group are closely related to the notion of Schwartz operators defined in [41].

COMMENT # 1.4

*p. 3: “This is a slightly weaker property than energy limited quantum dynamics”. Is it not a slightly *stronger* property, in the sense that any evolution that satisfies (2) is also energy limited, but not vice versa?*

Reply:

Let us define “energy limited dynamics” as a channel that sends *all finite energy states* to states of finite energy (this is one of the equivalent definitions in arxiv:2405.10259, Ref. [30] in the main text. Then our definition of “physical” is weaker because it only requires the output to be energy-bounded *when the input is a finite superposition of Fock states*. If a channel is physical in the sense of Eq. 2 then it might still send some finite-energy input state to a state of infinite energy, provided this input state has infinite support on the Fock basis. On the other hand, if a channel is energy-limited, it will send any finite-energy input to a finite-energy output. In particular it will send any finite superposition of Fock states to some finite energy state.

COMMENT # 1.5

Ref. [32] has a published version, which should be cited.

Reply:

We thank the referee for their comment, the published version is now cited as Ref. [32].

COMMENT # 1.6

For an important result such as the gentle measurement lemma, a quick citation to a textbook such as [41] is in my view not appropriate. One should spend some time to trace back the contribution to the original works, which in this case are Lemma 4.3 in E. B. Davies. Quantum stochastic processes, Communications in Mathematical Physics 15(4), 277-304 (1969), whose proof contains essentially the statement, even in its tight form, as well as the better known A. Winter. Coding theorem and strong converse for quantum channels, IEEE Transactions on Information Theory 45(7), 2481-2485 (1999).

Reply:

We thank the referee for their comment, we added pointers to the suggested references below Lemma 1. In the same place, we also mentioned the tighter result in <https://arxiv.org/abs/2501.12447>, as mentioned by the referee in Comment 1.9 below.

Changes:

The added paragraph reads:

Earlier proofs of essentially the same result can be found in [42,43]. Furthermore, while we will use this version for the sake of simplicity, a slightly tighter bound was recently proven [44] (see Lemma 6 therein).[FA: check numbers after final biblio modifications]

COMMENT # 1.7

“notation” is uncountable; thus, “notations” does not exist in English.

Reply:

We thank the referee for spotting this mistake, we corrected it.

COMMENT # 1.8

SI, p.2: “we now look for \hat{V} such that $\|(\mathcal{U} - \mathcal{V})(\phi)\|_1$ is uniformly bounded”. It is for sure uniformly bounded, by the constant 2.

Reply:

We made the statement more precise as in the following changes.

Changes:

we now look for \hat{V} such that $\|(\mathcal{U} - \mathcal{V})(\phi)\|_1$ is uniformly bounded by a suitably small constant.

COMMENT # 1.9

The authors might be interested in knowing that Lemma 1 is not tight. A tight version has been recently proposed, under the name “gentler measurement lemma” as Lemma 6 (and, more precisely, Eq. (44)) in <https://arxiv.org/abs/2501.12447>.

Reply:

See reply to comment 1.6 above.

COMMENT # 1.10

Lemma 2 in the SI is of course part of the folklore of quantum information, although I have to admit that a quick search did not reveal any precise pointers. A one-line proof uses the formula $\|X\|_1^2 = 2\text{Tr}[X^2] - (\text{Tr}[X])^2$, valid for any 2×2 matrix X with one positive and one negative eigenvalue.

Reply:

Indeed, the proof suggested by the referee is essentially a more compact version of our proof. Our proof is recovered once the r.h.s. of the matrix equation above is expanded. We included this well known result for reference and we will keep our original version since we prefer a more explicit phrasing.

COMMENT # 1.11

A general comment: I recall that a discussion on the open question tackled in this work took place at the conference "Towards Infinite Dimension and Beyond in Quantum Information", held in Granada in May 2023. It would have been nice to acknowledge the contribution of the attendees of that conference for the development of the present work, if any.

Reply:

We thank the referee for their suggestion which we have followed in the revised version.

Changes:

The added sentence reads:

U.C. is grateful to Á. Capel, N. Datta, L. Lami and A. Winter for organising the BIRS-IMAG workshop Towards infinite dimension and beyond in quantum information (May 2024, Granada, Spain), where open questions resolved in this work were discussed.

COMMENT # 1.12

p.5, bottom of 1st column: there are quite a few repetitions that diminish the quality of the prose; my tally is engineer(...) x3 and simulat(...) x2.

Reply:

We thank the referee for pointing this out. We have modified the paragraph as in the following changes.

Changes:

Beyond quantum state ~~engineering preparation~~, Theorem 2 provides a universal method for ~~engineering exactly exactly reproducing~~ any finite-dimensional Hamiltonian using polynomial Hamiltonians. This method can be used to engineer universal bosonic simulators capable of ~~simulating emulating~~ the unitary evolution of any ~~finite-dimensional discrete-variable~~ quantum system.

REVIEWER # 2

COMMENT # 2.1

Please find my report attached in the pdf file. At this point, I recommend acceptance after minor revision, as this paper resolves key conceptual and technical questions in CV quantum theory in very precise yet conceptually clear manner.

Reply:

We are grateful to the referee for their positive assessment of our work.

COMMENT # 2.2

This manuscript addresses long-standing questions regarding the validity and completeness of effective descriptions of bosonic quantum systems; specifically, (1) effective dimension truncations and (2) effective Hamiltonians (restricted to polynomials in canonical operators). These approximations underpin much of continuous-variable (CV) quantum information theory, CV quantum computing, and simulation (of bosonic systems and of finite-dimensional systems with bosonic simulators). The authors resolve conceptual and technical questions surrounding whether these approximations can faithfully capture the dynamics of bosonic systems (in a positive way). Their contributions are organized around four main theorems, with Theorems 1 and 2 constituting the foundational technical results:

- Theorem 1: Shows that any physical unitary evolution can be approximated to arbitrary precision by a finite-dimensional unitary, with explicit control over the approximation in terms of an energy- constrained diamond norm. The effective dimension needed for approximation can be computed explicitly.*
- Theorem 2: Demonstrates that any finite-dimensional Hamiltonian can be exactly reproduced by a polynomial Hamiltonian over the bosonic mode. This is achieved constructively using Lagrange interpolation polynomials.*

Together, these results show that any physical bosonic unitary can be approximated arbitrarily well using only polynomial Hamiltonians, establishing the soundness of the Lloyd–Braunstein

model of CV universality. This leads to, e.g., (i) bosonic Solovay–Kitaev-type theorem (Theorem 4); (ii) a constructive foundation for universal bosonic quantum simulators capable of modeling finite-dimensional systems (e.g., qudits) using CV architectures.

Strengths

- The manuscript addresses foundational open problems and provides rigorous, constructive proofs.
- The results have broad relevance for CV quantum computing, bosonic simulation, and quantum information theory.
- The authors provide conceptual clarity about why the use of polynomial Hamiltonians and Hilbert space truncations are not merely heuristic—but are in fact complete and universal in a well-defined operational sense.

Reply:

We are grateful to the referee for taking the time to read our manuscript carefully.

COMMENT # 2.3

I only have 1 direction in terms of critiques, regarding the effective dimension d , scaling, and efficiency of simulations. The effective dimension d scales as

$$d = \mathcal{O} \left(\frac{E}{\epsilon^4} E_U \left(\frac{64E}{\epsilon^2} \right) \right),$$

where $E_U(n)$ denotes a supremum of the energy after evolution over a particle-number-truncated subspace.

Reply:

Indeed, this is a nuanced point. We clarify in the replies below how the energy added to the system really needs to be evaluated case by case, and that the complexity depends on the specific evolution U .

COMMENT # 2.4

Can the authors clarify the growth behavior of $E_U(n)$? Are there bounds or examples illustrating its scaling? The authors comment at the end of paragraph that it “...may grow very fast...” But that’s all.

Reply:

As it turns out, the energy growth can be arbitrarily fast, as demonstrated by the following explicit construction:

Recall the definition of the maximal amount of energy involved when implementing a physical unitary evolution $\hat{U} \in \mathcal{U}(\mathcal{S})$ of an initial state with a number of particles at most n :

$$E_{\hat{U}}(n) := \sup_{|\psi\rangle \in \mathcal{H}_n} \langle \psi | \hat{U}^\dagger \hat{n} \hat{U} | \psi \rangle < +\infty. \quad (1)$$

We give an explicit construction of physical unitary channels for which this quantity can grow arbitrarily fast with photon number:

1. Pick some sparse countably-infinite subset X of \mathbb{N} , e.g., the multiples of 10 excluding zero.
2. Pick some bijection $g : \mathbb{N} \setminus X \rightarrow \mathbb{N} \setminus 2^{(X)}$ (which are both countably infinite sets)
3. Now, define a unitary \hat{U} acting on Fock states as

$$\hat{U}|x\rangle = \begin{cases} |2^x\rangle, & \text{if } x \in X \\ |g(x)\rangle, & \text{if } x \in \mathbb{N} \setminus X \end{cases} \quad (2)$$

Here, \hat{U} is physical by construction, because it sends Fock states to Fock states, and $E_{\hat{U}}(n)$ grows at least as fast as 2^n . This reasoning is not specific to the exponential function but can be repeated for any function defined on \mathbb{N} , showing that there are physical unitaries \hat{U} for which $E_{\hat{U}}(n)$ grows arbitrarily fast in n .

Changes:

This construction was added as a new section in the Supplementary Information, and referred to in the main text after Theorem 1:

We give an explicit construction in the Supplementary Information showing that this growth may be arbitrarily fast.

COMMENT # 2.5

Is the overall scaling tight, or could it be improved in restricted settings (e.g., Gaussian evolutions)?

Reply:

As anticipated by the referee, the scaling of d in Theorem 1 is indeed not tight, due to the overhead coming from the gentle measurement lemma used in the proof which is not always necessary. In particular, for unitary channels which preserve the particle

number, such as (Gaussian) phase-shifters, the step of the proof in which the unitary is being truncated in Fock basis is not necessary, since the particle-number-preserving channels are already block-diagonal in the Fock basis. We expect that a case-by-case analysis could significantly tighten the bound on the overall scaling.

COMMENT # 2.6

Are the authors aware of specific (rather than typical) regimes where this scaling becomes unmanageable?

Reply:

The unitary channels described in the reply to Comment 2.4 provide such a specific example where the fast scaling in energy leads to an unmanageable scaling in effective dimension.

COMMENT # 2.7

Comment: Perhaps some of this is addressed qualitatively (or posed as open problems) in the outlook section. In particular, the statement regarding poor scaling of complexity with qudit dimension (i.e., effective dimension d)

Reply:

We believe that the unitary channels described in the reply to Comment 2.4 and added as a new section in the revised Supplementary Information clarify this point.

Note that the scaling of complexity with qudit dimension mentioned in the discussion refers to an additional cost incurred by the use of the Solovay–Kitaev theorem, corresponding to the effective dimension, itself scaling with the energy.

COMMENT # 2.8

I recommend acceptance after minor revision. This paper resolves key conceptual and technical questions in CV quantum theory.

Reply:

We are grateful to the referee for appreciating our work and advising for its publication.

Referee Report on “Can effective descriptions of bosonic systems be considered complete?”

June 7, 2025

Summary and Contributions

This manuscript addresses long-standing questions regarding the validity and completeness of **effective descriptions** of bosonic quantum systems; specifically, (1) *effective dimension truncations* and (2) *effective Hamiltonians* (restricted to polynomials in canonical operators). These approximations underpin much of continuous-variable (CV) quantum information theory, CV quantum computing, and simulation (of bosonic systems and of finite-dimensional systems with bosonic simulators).

The authors resolve conceptual and technical questions surrounding whether these approximations can faithfully capture the dynamics of bosonic systems (in a positive way). Their contributions are organized around four main theorems, with **Theorems 1 and 2** constituting the foundational technical results:

- **Theorem 1:** Shows that *any* physical unitary evolution can be approximated to arbitrary precision by a *finite-dimensional* unitary, with explicit control over the approximation in terms of an energy-constrained diamond norm. The effective dimension needed for approximation can be computed explicitly.
- **Theorem 2:** Demonstrates that *any* finite-dimensional Hamiltonian can be *exactly* reproduced by a *polynomial Hamiltonian* over the bosonic mode. This is achieved constructively using Lagrange interpolation polynomials.

Together, these results show that **any physical bosonic unitary can be approximated arbitrarily well using only polynomial Hamiltonians**, establishing the soundness of the Lloyd–Braunstein model of CV universality. This leads to, e.g., (i) **bosonic Solovay–Kitaev-type theorem** (Theorem 4); (ii) a constructive foundation for **universal bosonic quantum simulators** capable of modeling finite-dimensional systems (e.g., qudits) using CV architectures.

Strengths

- The manuscript addresses foundational open problems and provides rigorous, constructive proofs.
- The results have broad relevance for CV quantum computing, bosonic simulation, and quantum information theory.
- The authors provide conceptual clarity about why the use of polynomial Hamiltonians and Hilbert space truncations are not merely heuristic—but are in fact complete and universal in a well-defined operational sense.

Critiques and Questions

I only have 1 direction in terms of critiques, regarding the effective dimension d , scaling, and efficiency of simulations.

The effective dimension d scales as

$$d = \mathcal{O} \left(\frac{E}{\epsilon^4} E_{\hat{U}} \left(\frac{64E}{\epsilon^2} \right) \right),$$

where $E_{\hat{U}}(n)$ denotes a supremum of the energy after evolution over a particle-number-truncated subspace.

- Can the authors clarify the growth behavior of $E_{\hat{U}}(n)$? Are there bounds or examples illustrating its scaling? The authors comment at the end of paragraph that it “...may grow very fast...” But that’s all.
- Is the overall scaling tight, or could it be improved in restricted settings (e.g., Gaussian evolutions)?
- Are the authors aware of specific (rather than typical) regimes where this scaling becomes unmanageable?
- Comment: Perhaps some of this is addressed qualitatively (or posed as open problems) in the outlook section. In particular, the statement regarding poor scaling of complexity with qudit dimension (i.e., effective dimension d).

Recommendation

I recommend acceptance after minor revision. This paper resolves key conceptual and technical questions in CV quantum theory.